# Biomechanical Evaluation of Promising Different Bone Substitutes in a Clinically Relevant Test Set-Up

**DOI:** 10.3390/ma12091364

**Published:** 2019-04-26

**Authors:** Theresa Brueckner, Philipp Heilig, Martin Cornelius Jordan, Mila Marie Paul, Torsten Blunk, Rainer Heribert Meffert, Uwe Gbureck, Stefanie Hoelscher-Doht

**Affiliations:** 1Department for Functional Materials in Medicine and Dentistry, University of Wuerzburg, Pleicherwall 2, 97070 Wuerzburg, Germany; brueckner.theresa@gmail.com (T.B.); uwe.gbureck@fmz.uni-wuerzburg.de (U.G.); 2Department of Trauma, Hand, Plastic and Reconstructive Surgery, University Clinics of Wuerzburg, Oberduerrbacher Strasse 6, 97080 Wuerzburg, Germany; heilig_p@ukw.de (P.H.); jordan_m@ukw.de (M.C.J.); mila.paul@uni-wuerzburg.de (M.M.P.); blunk_t@ukw.de (T.B.); meffert_r@ukw.de (R.H.M.)

**Keywords:** drillable, cement, bone substitute, struvite, magnesium, biomechanical test, tibial fracture, apatite, biomaterials

## Abstract

(1) Background: Bone substitutes are essential in orthopaedic surgery to fill up large bone defects. Thus, the aim of the study was to compare diverse bone fillers biomechanically to each other in a clinical-relevant test set-up and to detect differences in stability and handling for clinical use. (2) Methods: This study combined compressive strength tests and screw pullout-tests with dynamic tests of bone substitutes in a clinical-relevant biomechanical fracture model. Beyond well-established bone fillers (ChronOS^TM^ Inject and Graftys^®^ Quickset), two newly designed bone substitutes, a magnesium phosphate cement (MPC) and a drillable hydrogel reinforced calcium phosphate cement (CPC), were investigated. (3) Results: The drillable CPC revealed a comparable displacement of the fracture and maximum load to its commercial counterpart (Graftys^®^ Quickset) in the clinically relevant biomechanical model, even though compressive strength and screw pullout force were higher using Graftys^®^. (4) Conclusions: The in-house-prepared cement allowed unproblematic drilling after replenishment without a negative influence on the stability. A new, promising bone substitute is the MPC, which showed the best overall results of all four cement types in the pure material tests (highest compressive strength and screw pullout force) as well as in the clinically relevant fracture model (lowest displacement and highest maximum load). The low viscosity enabled a very effective interdigitation to the spongiosa and a complete filling up of the defect, resulting in this demonstrated high stability. In conclusion, the two in-house-developed bone fillers revealed overall good results and are budding new developments for clinical use.

## 1. Introduction

In orthopedic surgery, bone defects occur after fractures or in tumor operations. To fill up major bone defects, bone substitutes are often needed, especially in cases when autologous bone grafting is insufficiently available. Tibial head depression fractures, typical fractures of older patients, are common examples of daily clinical practice in which a metaphyseal bone defect regularly remains after fracture reduction [1]. The availability and application of autologous bone grafts as the only osteogenic, osteoinductive, and osteoconductive material is limited [2,3]. Allogenous or xenogenous grafts have reduced osteogenic and osteoinductive properties as well as a reduced mechanical strength due to sterilization [3,4]. Thus, currently used alternatives are based on synthetic bone substitutes, such as self-setting calcium phosphate cements (CPCs). Though CPCs possess only osteoconductive properties [5], they avoid the downsides associated with conventional bone grafts and provide injectability ensuring a proper filling of bone defects. Depending on the end product of the setting reaction, two major CPC types are differentiated. If the pH value during setting is below 4.2, brushite, a metastable calcium orthophosphate is formed, whereas a pH value above 4.2 produces hydroxyapatite, which has a similar composition and crystallinity to the inorganic phase of the mineral bone matrix [5]. While brushite cements are known to be degradable in the human organism within months by both passive dissolution and acid producing osteoclasts [6], hydroxyapatite cement (HA) is presumably remodeled similar to human bone, only within years, by solely osteoclastic activity due to the thermodynamic stable character of HA under physiological conditions [7,8]. 

Comparative biomechanical studies and adequate treatment recommendations for the clinical application of the available bone cements are missing. Moreover, no drillable bone substitute which would ensure a complete filling of the defect is currently available on the market [9]. In a clinical context, drillability of mineral bone cements is defined as the possibility to drill a hole and insert a bone screw into the partially or fully set cement without fracturing the cement implant during processing. Additionally, if an apatite cement is applied, the resorbing and reorganization back to trabecular bone may not take place, thereby affecting long-term stability [10]. 

To improve the drillability and biodegradability of bone substitutes, two new approaches have been developed in the last years. CPCs gained drillable characteristics either by modification with fibers, as it was realized in the former CPC Norian^®^ Drillable, or through the incorporation of reactive monomers, a concept of dual setting cement systems firstly described by Dos Santos et al. in 1999 [11]. The in situ polymerization of those monomers forming an interpenetrating network of ceramic matrix and hydrogel is known to introduce a pseudoplastic mechanical fracture behavior [12]. Moreover, biodegradability of cements can be obtained by using struvite-forming magnesium phosphate cements (MPCs), with an assumed degradation within 10–12 months in vivo [13] as a result of their high solubility (pKs = 12–14) [14]. Also, MPCs exhibit high initial compressive strength values above 50 MPa [15,16,17,18,19].

So far, however, a bone substitute combining drillability, high biomechanical strength, and resorption in vivo is not commercially available. Therefore, this study analyzed two new promising bone substitutes: An in-house-developed dual setting calcium-deficient hydroxyapatite (Ca_9_HPO_4_(PO_4_)_5_OH) cement [20] gaining the clinical desired drillability from an incorporated [21] poly-2-hydroxyethyl methacrylate (HEMA)-hydrogel and an in-house-developed struvite (NH_4_MgPO_4_·6H_2_O)-forming MPC [17]. Cements often used in clinical contexts are the brushite cement (CaHPO_4_·2H_2_O) ChronOS^TM^ Inject and the apatite cement Graftys^®^ Quickset. ChronOS^TM^ inject is known to have a good degradation rate (within 18 months, manufacturer information) without a lack of strength in this time [6] and the struvite cement has been demonstrated to show quantitative bone remodeling in an large animal model within ten months for both unloaded and load-bearing defects [22,23]. Such struvite cements provide a high initial strength of up to 80 MPa under compression [22], while ChronOS™ inject is mechanically much weaker with a compressive strength of <1 MPa. In contrast, both the in-house-developed hydroxyapatite cement and Graftys^®^ Quickset are likely only slow-degrading materials due to their low soluble hydroxyapatite matrix. The latter offers a good primary stability (24 MPa after 24 h, manufacturer information) and the HEMA-modified hydroxyapatite cement shows compressive strengths of ~30 MPa after 7 days setting [20]. 

The aim of the study was to detect biomechanical differences in stability and handling for clinical use of four injectable bone substitutes in a clinically relevant test set-up. Thus, in addition to a basic biomechanical investigation, this study newly analyzed the interaction of the bone substitutes in a fracture model in tibial head depression fractures. It was hypothesized that the in-house-prepared bone cements would provide an equivalent biomechanical stability as the commercial counterparts combined with additional advantages for the clinical application, such as the ability to be drilled after a short period of setting due to their pseudo-plastic behavior and their high strength.

## 2. Materials and Methods 

### 2.1. Raw Powder and Cement Preparation of In-House-Developed Bone Cements

α-tricalcium phosphate (α-TCP, Ca_3_(PO_4_)_2_) was synthesized by sintering a 2.15:1 molar ratio of CaHPO_4_ (J.T. Baker, Griesheim, Germany) and CaCO_3_ (Merck, Darmstadt, Germany) for 5 h at 1400 °C in a sintering furnace (Oyten Thermotechnic, Oyten, Germany). Farringtonite (Mg_3_(PO_4_)_2_) was synthesized sintering 0.6 mol MgHPO_4_∙3H_2_O (Sigma Aldrich, Steinheim, Germany) with 0.3 mol Mg(OH)_2_ (Sigma Aldrich, Steinheim, Germany) for 5 h at 1050 °C. The sintering cakes of both systems were crushed and sieved <355 µm and milled for 1 h (farringtonite) or 4 h (α-TCP) in a planetary ball mill (PM400, Retsch, Haan, Germany).

In case of the dual setting apatite formulation, α-TCP was mixed with 0.5% ammoniumpersulfate (Sigma Aldrich, Steinheim, Germany). The liquid phase of the cement paste was composed of 50% HEMA (Sigma Aldrich, Steinheim, Germany), 2.5% Na_2_HPO_4_ (Merck, Darmstadt, Germany), and 0.25% *N*,*N*,*N*′,*N*′ tetramethylethylene diamine (Sigma Aldrich, Steinheim, Germany). Both the solid and the liquid phase of the cement paste were homogenously mixed by means of a spatula and glass slab for 30 s at a powder-to-liquid ratio (PLR) of 1.6 g/mL. For the in-house-prepared struvite cement system, the farringtonite raw powder was equally mixed homogeneously for 30 s with an aqueous 3.5 M (NH_4_)_2_HPO_4_ (Sigma Aldrich, Steinheim, Germany) solution at a PLR of 2.0 g/mL.

The commercially available cement compounds, ChronOS^TM^ Inject and Graftys^®^ Quickset, form brushite and calcium-deficient hydroxyapatite, respectively, and were purchased from Depuy Synthes (Umkirch, Germany) and Graftys (Aix-en-Provence, France), respectively. They were prepared according to the manufacturers’ instructions. All used cement compositions are summarized in Table 1.

### 2.2. X-ray Diffraction and Mercury Porosity Analysis

X-ray diffraction patterns of the in-house-prepared cement raw powders and the set cements were recorded with a Siemens D5005 diffractometer (Siemens, Karlsruhe, Germany) in a 2θ range from 20°–40° with a step size of 0.01°. The qualitative phase composition was evaluated by comparing the diffraction patterns with reference patterns from the PDF-database. Mercury porosity analysis (PASCAL 140/440, Porotec GmbH, Hofheim, Germany) was applied to compare relative and cumulative pore volume, average and median pore diameter, and visualize pore size distribution. Therefore 250–350 mg of each sample was placed in a dilatometer and filled with mercury. At an applied pressure of up to 400 MPa, the pore size distribution in the range of macropores and mesopores (0.01–10 μm) was determined. For evaluation, the software “solid” was used.

### 2.3. Static Mechanical Testing: Compressive Strength and Screw Pullout Test Setup

The as-prepared cement pastes were filled in cuboid silicone rubber molds with an aspect ratio of 2:1 (12 mm × 6 mm × 6 mm) and the test blocks (at least n = 10 per group) were stored in distilled water at 37 °C for 24 h before testing. Wet samples were tested in axial compression at a crosshead speed of 1 mm/min by means of the universal testing machine Z020 (Zwick, Ulm, Germany). The compressive strength was then calculated by dividing the maximum load at failure by the cross-sectional area of the test blocks.

For testing the screw pullout force, a similar test set-up to former biomechanical studies was used [24,25]: Silicone rubber molds were used to prepare cylindrical cement samples of 20 mm in length and 15 mm in diameter. Through a hole in the bottom, cortical screws (25 mm × 3.5 mm, DePuy Synthes, Umkirch, Germany) were put in the molds before cement was filled in, enabling a standardized screw embedding depth of 15 mm. For drilling screws into the samples, we used molds with a central notch at the bottom, serving as a mark on the pre-set cement for the subsequent drilling, tapping, and screwing (Figure 1a). To ensure the same depth of penetration, a 2.5 mm drill and a 3.5 mm tap and screw were marked 15 mm from the apex. Before testing, the samples (n = 10 per group) were placed for 24 h in distilled water at 37 °C. Every sample was examined by X-ray to detect air bubbles, ensure straight screw placement and equal embedment depth (Figure 1b). For every bone substitute, one group with embedded screws was tested. In addition, one group with manually placed screws was tested for the drillable dual setting CPC. An axial tensile test was performed at a speed of 1 mm/min through the traverse and tubular holder moving upwards to determine the screw pullout force (Figure 1c).

### 2.4. Cyclic Biomechanical Testing within a Tibial Depression Fracture Model

To generate pure depression fractures of the lateral tibial plateau (AO 41-B2.2, Schatzker III), a validated fracture model in synthetic bones (Synbone^®^ 1110, Synbone, Malans, Switzerland) [9,26], was chosen. Consequently, tibiae were cut at mid-diaphysis 20 cm below the tibial plateau and embedded with gypsum at 5° valgus in a custom-made device [27]. Five predetermined breaking points were set with a 2 mm drill in a 12 mm circle centrally on the lateral plateau (Figure 2a). Subsequently, the apparatus was mounted on the universal testing machine, the 12 mm indenter was positioned exactly over the breaking points, and an axial load was applied with 500 mm/min to produce a 15 mm-deep pure depression fracture (Figure 2b,c).

Fractures were reduced indirectly by the often-used clinical technique ARIF (arthroscopically supported reduction and internal fixation) [9,26,28,29]. Through a metaphyseal cortical window, a K-wire was placed under the depressed articular fracture fragment (Figure 2d), a tunnel was drilled (Figure 2e) and a K-wire-guided cannulated ram (Figure 2f) was then inserted to compact the subchondral spongiosa and thus restore the plateau anatomically to a plane joint surface (Figure 2g). In groups 1 to 3 the metaphyseal void remaining after reduction was filled with bone substitute only (Figure 3a,b). Specimens were stored dry at 37 °C for 24 h in an incubator to allow comparison with previous studies [9,26,28]. In contrast, all residual groups were stored with distilled water-soaked gauzes wrapped around the bone at 37 °C for 24 h [27,30] (Table 2). This storage takes the high water demand of the HEMA-hydrogel of the CPC into account and enables post-curing of the ceramic constituent [20], allowing further comparability. In groups 5 and 6, the fracture was stabilized with a four screw “jail technique” after reduction [9,26,31]. Two 3.5 mm cortical bone screws were inserted by two anterior small incisions and two 6.5 mm cancellous bone screws by lateral incisions, supporting the reduced fragment like a grid (Figure 2h and Figure 3d). The remaining metaphyseal void was then filled up retrogradely with bone substitute injected through a cannula (Figure 3c, Table 2). This procedure was done backwards for group 7, in which the defect was filled up first with drillable CPC and then drilled after 10 min of pre-setting (Table 2). To assess the filling of the metaphyseal defect and the position of the screw osteosynthesis, specimens were examined by X-rays before storage (Figure 3b,d). 

For testing, the specimens were fixed in the universal testing machine Z020 (Zwick Roell, Ulm, Germany), the indenter was positioned exactly over the reduced fracture fragment, and an axial load was applied (Figure 3e). The testing protocol included a cyclic loading phase followed by maximal loading and has been validated in previous studies [9,26,28]. Ten settling cycles from 20 to 125 N were followed by 3000 measuring cycles from 20 to 250 N with 25 mm/min [27]. The force levels in the protocol were chosen in compliance with the postoperative partial weight bearing conditions of around 20 kg of the operated limb [29] and the number of cycles (3000) has been found to be sufficient for detecting differences in displacement [9,26,28]. After the last measuring cycle, axial load was steadily increased with a constant speed of 100 mm/min until failure (Figure 4).

The parameters of interest were the displacement of the reduced fracture fragment in the cyclic loading phase, the maximum load, and the stiffness of the load-to-failure tests.

Compressive strength and maximum pullout force (Section 2.3), as well as the displacement and maximum load in the fracture model biomechanical test set-up (Section 2.4), were recorded by a 20 kN load cell at the traverse of the universal material testing machine, Zwick Roell Z020, Ulm, Germany. Additionally, the pullout stiffness was calculated as the slope during the elastic deformation curve in the load-displacement diagrams, either in the screw pullout tests (Section 2.3) or in the load-to-failure tests after the cyclic loading phase (Section 2.4).

### 2.5. Stereomicroscopic and Scanning Electron Microscopy Images

To analyze the interdigitation of the bone cement into the adjacent spongiosa, specimens of the dry and humid CPC and MPC were prepared and cut sagittally at the lateral tibial plateau through the filled-up bone defect. Subsequently, images of the exposed surface were taken with a stereomicroscope (Carl Zeiss, Oberkochen, Germany). The aforementioned specimens were then further prepared with a diamond saw and dried for 6 days in a desiccator (Pfeiffer Vacuum, Aßlar, Germany). After sputter coating with a 4 nm layer of platinum (Leica EM ACE 600, Leica Microsystems, Wetzlar, Germany) the interface between bone cement and spongiosa was analyzed by scanning electron microscopy (Crossbeam 340, Carl Zeiss, Oberkochen, Germany) with an acceleration voltage of 3.0 kV via detection of secondary electrons. 

### 2.6. Statistical Analysis

The number of specimens for the experimental groups (n = 9) was estimated by power analysis using a significance level of 5% and a power of 80%. The calculation of effect size *d* was based on the results of a comparable pilot study. Descriptive statistics (means and standard deviations) for the outcome variables were initially calculated for each of the experimental groups (expert’s report by the statistical institute of the mathematical department, University of Wuerzburg, Germany).

Normal distribution was confirmed and significant differences were calculated by one-way ANOVA. Non-normally distributed data were analyzed by a Kruskal–Wallis test, followed by a Mann–Whitney U-test to find significant differences between groups. The statistical analyses were conducted using IBM^®^ SPSS^®^ Statistics 21, with the level of significance set at *p* < 0.05.

## 3. Results

### 3.1. Composition and Porosity Analysis of Cement Samples

The in-house-prepared magnesium phosphate cement consisted of farringtonite (Figure 5A), which formed struvite after the reaction with ammonium phosphate from the cement liquid (Figure 5B). Due to non-stoichometric mixing ratios between cement powder and liquid, the reaction was not quantitative and a large portion of farringtonite remained unreacted in the set cement. The dual setting α-TCP/HEMA cement consisted of pure α-tricalcium phosphate (Figure 5C), which formed a nanocrystalline hydroxyapatite matrix after setting (Figure 5D). Mercury porosity analysis (Figure 6) showed the highest cumulative pore volume of 427 mm^3^/g, an average pore diameter of 16 nm and a median pore diameter of 20 nm for commercially available CPC, Graftys^®^ Quickset. Cumulative values were slightly lower for ChronOS^TM^ Inject with 397 mm^3^/g, but pore size distribution was different, with an average pore diameter of 544 nm and a median pore diameter of 1000 nm. The two in-house formulations presented the lowest cumulative pore volumes, with 67 mm^3^/g, 88 nm, and 224 nm, respectively, for the MPC and 38 mm^3^/g, 123 nm, and 306 nm, respectively, for the drillable CPC.

### 3.2. Static Mechanical Testing: Compressive Strength and Screw Pullout Tests

The compressive strength after 24 h of setting in water at 37 °C is revealed by Figure 7. With a value of 0.6 ± 0.1 MPa, the lowest compressive strength was observed by commercially available brushite-forming cement (ChronOS^TM^ Inject), while the struvite-forming in-house-prepared MPC actually showed a compressive strength of approximately 100 MPa. Though both residual groups form calcium-deficient hydroxyapatite as the main mineral setting product with a compressive strength of 19.0 ± 2.5 MPa, the commercial version (Graftys^®^ Quickset) was almost three times as strong compared to the in-house-developed formulation. All differences between the single groups were significant, with p < 0.01.

Only the in-house-prepared dual setting CPC sustained tapping and drilling after a short pre-setting time of 10 min due to its pseudoplastic fracture behavior, as previously shown for cements with similar compositions [12]. Thus, for the other cement groups, the cortical screws were solely embedded in the cement matrix prior to setting. As shown by Figure 8a, the maximum pullout forces each exhibited similar tendencies as already proved within the compression test setup for the pure cement specimens (Figure 7). Mean maximum pullout forces lay between 41 ± 7 (ChronOS^TM^ Inject) and 1704 ± 248 N (magnesium phosphate), while all differences between the single groups were significant (*p* < 0.01). When choosing embedment instead of manual drilling in the case of the dual setting CPC, a 2.3-fold increase of the maximum pullout force from 129 ± 38 to 295 ± 39 N occurred (Figure 8a).

Analyzing the ascending slopes of corresponding force-displacement curves revealed relevant stiffness data for the five different groups (Figure 8b). With the exception of the commercial hydroxyapatite formulation (Graftys^®^ Quickset), the stiffness of the materials slightly, but not significantly, increased in accordance with the previously reported maximum pullout force from commercial brushite (ChronOS^TM^ Inject, 534 ± 258 N/mm) over in-house-prepared dual setting hydroxyapatite cement (calcium phosphate, 799 ± 258 N/mm) to in-house-prepared struvite cement (magnesium phosphate, 1089 ± 408 N/mm). Equally, a slight decrease in stiffness by approximately 20% was observed when manually drilling the material instead of embedding the cortical screws (dual setting calcium phosphate). With a stiffness of more than 3000 N/mm, the commercial hydroxyapatite-forming Graftys^®^ Quickset exhibited the significantly highest value of all experimental groups (Figure 8b).

### 3.3. Dynamic Biomechanical Testing within a Tibial Depression Fracture Model

ChronOS^TM^ Inject, the commercially available brushite-forming cement formulation, exhibited compressive strength and screw pullout force, which were at least one order of magnitude lower than the residual formulations. Thus, it was not examined further by the elaborate dynamic tests in the tibial head depression fracture model, of which Figure 9 depicts the results. Obviously, the lowest overall displacement of ~1.5 mm occurred for augmenting the depression fracture model with MPC, both with cement only as well as applying the jail technique, but only in the case of pure augmentation was the difference toward other cements significant, at *p* < 0.01. In general, both storage conditions (dry and humid) and the combination with screw osteosynthesis did not seem to have an impact on the displacement; however, a slight decrease of approximately 0.7 mm from 2.5 ± 0.4 to 1.8 ± 0.6 mm was observed in the case of the dual setting calcium phosphate with the jail technique in contrast to without it (Figure 9a). The tendencies, when only considering the measuring cycles without the settling cycles (Figure 9b), were quite similar, though the displacement caused by the measuring cycles only made up less than the half of the total displacement. Again, the in-house-prepared magnesium phosphate exhibited the lowest displacement of 0.6 ± 0.1 mm without osteosynthesis, but the difference between magnesium phosphate and Graftys^®^ Quickset did not reach statistical significance beyond this and the difference toward other formulations was equally marginal when applying the jail technique (Figure 9b).

As seen in Figure 9c, the additional osteosynthesis improved the mechanical outcome significantly of the treated tibial depression fractures for every type of bone substitute, such that the maximum load before failure increased for example from >1.4 kN (calcium phosphate only) to up to 3.75 ± 0.31 kN (calcium phosphate with the jail technique). Under humid storage conditions, calcium phosphate exhibited a significantly higher maximum load compared to the samples stored under dry conditions with *p* < 0.01. 

Regarding the corresponding stiffness values, similar tendencies as for the maximum load were observed for the different cement systems, i.e., the lowest stiffness occurred for the use of the in-house-prepared dual setting calcium phosphate (approximately 250–290 N/mm for pure bone substitute; 430 ± 66 N/mm for the combination with screws) and the highest stiffness in the case of in-house-prepared magnesium phosphate (477 ± 52 N/mm for pure bone substitute; 661 ± 62 N/mm for jail technique). For all three cement formulations, the commercial Graftys^®^ Quickset and the in-house-prepared calcium and magnesium phosphate cements, the stiffness was significantly higher with the additional fixation through jail technique. The highest increase in stiffness, approximately 50%, revealed the calcium phosphate cement in combination with screws compared to the screw-free control (humid conditions) (Figure 9d). In contrast to the results seen for the maximum load (Figure 9c), the storage conditions of in-house-prepared dual setting calcium phosphate did not have significant effects on the stiffness (Figure 9d).

### 3.4. Stereomicroscopic Images

Stored under dry conditions, a gap could be distinguished between the CPC cement body and the adjacent spongiosa (Figure 10a), whereas an improvement of the interdigitation was visible under humid storage (Figure 10b). Also, the MPC demonstrated a seamless interdigitation with complete filling of all proximate spongiosa cavities, including those directly under the articular fracture fragment (Figure 10c).

### 3.5. Scanning Electron Microscopy (SEM) Images

Consistent with the stereomicroscopic images (Figure 10c), SEM images confirmed the seamless interdigitation of the MPC with the adjacent spongiosa (Figure 11a), whereas for the experimental CPC, both after initial humid storage (Figure 11b) and after initial dry storage (Figure 11c), a gap between the cement body and the spongiosa could be distinguished. In the case of the CPC under initial humid storage, cement could also be detected in the nearby spongiosa cavities (arrowhead in Figure 11b). Both CPC samples were stored in a desiccator before performing SEM. A good interdigitation at the cement–spongiosa interface was also provided by Graftys^®^ Quickset (Figure 11d). Moreover, the macroporous and mesoporous structure of this bone cement became visible (arrowhead in Figure 11d).

## 4. Discussion

Difficulties with conventional bone cements, like a lack of drillability, uncertain resorbability [10], and mechanical weakness compared to human bone [32], emphasize the need for new alternative cements for clinical application. Therefore, two new concepts, i.e., a drillable apatite cement and a high strength, supposedly resorbable MPC, were mechanically and biomechanically evaluated against the clinically used formulations Graftys^®^ Quickset and ChronOS^TM^ Inject. Up until the date of the tests, both cements were often used in our clinic, although the brushite cement is actually not available anymore. 

Throughout static compressive strength testing, the in-house-prepared magnesium phosphate revealed the significant highest compressive strength compared to all other bone substitutes (Figure 7). Our findings correspond well with previously published studies, demonstrating early strength acquisition and high strength values (above 60 MPa) for MPCs [15,16,17]. The low porosity of the MPC (Figure 6a) might also play a role, as porosity and mechanical strength are inversely and exponentially linked [5]. In contrast to the results of this study, Christel et al. [20] demonstrated a greater than 4-fold higher compressive strength of 30 MPa for a similarly composed apatite cement, which is likely due to the differences in the PLR. Whereas Christel et al. [20] used a PLR of 3.0 g/mL, the lower PLR of 1.6 g/mL used here should ensure injectability of the cement paste. It is well described in literature that a higher PLR results in a higher compressive strength [5]. Moreover, the drillable apatite cement showed a pseudoplastic mechanical behavior, a phenomenon which is also described by the aforementioned publication, as the addition of HEMA resulted in a decrease of the bending modulus and a simultaneous increase of the work of fracture [20]. Both parameters enabled a similar cement formulation to be drilled after short pre-setting, which was shown here for the first time. The compressive strength of Graftys^®^ Quickset is well in accordance with the manufacturer’s data of 24 MPa and of a micro-, meso-, and macroporous structure, which could be confirmed by the mercury porosity analysis (Figure 6d) and SEM images (Figure 11d). ChronOS^TM^ Inject demonstrated a very low mechanical stability (Figure 7 and Figure 8), which is due to the fact that this cement was not cohesive when stored in distilled water. This observation has previously been reported by Luo et al. [33]. In addition, the porosity analysis showed the highest cumulative porosity for the brushite cement (Figure 6c).

Analogously, during screw pullout testing, the same tendencies were detected between the different bone substitutes. Further, it seemed that manually drilling and inserting the screw significantly weakened the screw–cement interface in comparison to that of embedded samples (Figure 8a). Chapman et al. [34] demonstrated that tapping a soft polyurethane foam increased the drill hole diameter by 27% and significantly reduced screw pullout strength. As the drillable apatite cement had a soft consistency after 10 min of pre-setting, the findings are in strong agreement with other authors, stating that tapping a soft material reduces pullout strength by setting a larger defect than the original drill hole [35,36]. This sounds especially conclusive regarding corresponding pullout stiffnesses of the dual setting bone substitutes. Those were comparable to specimens with ChronOS^TM^ Inject, which presumably softened in the aqueous environment due to its poor cohesiveness. Interestingly, samples with screw–cement combinations from Graftys^®^ Quickset revealed a three times higher stiffness than magnesium phosphate (Figure 8b), which may be explained by the fact that cements can behave different under compressive and tensile tests and, as a consequence, Graftys^®^ Quickset did not show stiffness values as anticipated from the tendencies of the compressive strength tests.

With respect to the settling and measuring cycles during the dynamical tests with bone substitute-treated tibial head depression fractures, the combination with MPC exhibited the significant lowest displacement in the groups without screws (Figure 9a). This might be explained by taking into consideration that complete filling of the proximate cancellous bone cavities was verified via stereomicroscopy (Figure 10) as well as via SEM (Figure 11a) and that the tested cement showed high primary stability in the pure material tests as well. Results proposed by Jordan et al. [28] support this explanation by finding a similar connection of significant lower displacement and higher stiffness with a better integration in the nearby synthetic spongiosa. The in-house-prepared drillable CPC revealed a high displacement when used as a bone substitute alone (Figure 9a), which was likely due to the aforementioned pseudoplastic characteristics. Such cement formulations enable the drilling of the screws after injection of the bone cement in order to optimize the filling precision of the paste also inside irregularly shaped defects. Therefore, the displacement of the drillable CPC could be significantly reduced in combination with the jail technique (Figure 9a). The issue with incomplete filling of non-drillable formulations could exemplarily be illustrated in the case of Graftys^®^ Quickset, where radiographs taken after fracture stabilization with screws disclosed that, in four of nine specimens, the screws worked detrimentally (data not shown). They hindered the complete filling of the defect to the subchondral area, which might account for the corresponding large displacement and high standard deviation (Figure 9a). Equally, Hoelscher-Doht et al. [9] previously demonstrated that using the jail technique with screw placement after replenishment resulted in a significant lower displacement and higher stiffness of the fixed tibial plateau depression fractures, as if the procedure was done in reverse. Overall, the displacement results of this study are in the range of the displacements measured in other studies for the same fracture type [9,26,28].

It is a consensus in the literature that there is a correlation between a remaining step after fracture reduction and post-traumatic arthritis [37,38]. Brown et al. [39] showed in an animal study that a fracture step in the cartilage of more than 1.5 mm leads to significantly higher stress than under physiological conditions. The obtained displacement of the reduced tibial fractures filled up with magnesium phosphate lies close to this value (Figure 9a). Honkonen et al. [40] demonstrated that the functional and clinical outcome was significantly deteriorated if a step-off of more than 3 mm was the case in tibial plateau fractures. What can be positively mentioned is that the displacements of all bone substitutes in this study were below 3 mm (Figure 9a).

As anticipated from the hygroscopic nature of the in-house-prepared drillable CPC, the storage conditions of the treated depression fracture models influenced the mechanical outcome such that samples with this cement exhibited a significantly higher load-bearing capacity when stored under humid conditions (Figure 9c). As already described, incorporation of the HEMA-hydrogel in the CPC changes its characteristics from brittle to pseudoplastic, going along with a decrease of the bending modulus and an increase of the work of fracture when stored in water. This theory might be confirmed by corresponding stereomicroscopic and SEM images illustrating a swelling of the whole cement (Figure 10a,b and Figure 11b,c). This presumably leads to a better interdigitation with the spongiosa and results in a higher maximum load (Figure 9c). Furthermore, such storage conditions allow for an additional post-hardening of the cement, which is not fully set even after 24 h, as demonstrated by X-ray diffraction analysis (Figure 5D), which can explain the higher load due to a higher degree of cement conversion. 

All groups with an additional osteosynthesis provided a significantly higher maximum load compared to the groups in which only cement was used (Figure 9c). This again confirms the results of previous studies that display the mandatory nature of osteosynthesis for the maximum load-bearing capacity of the fixed fracture [26,28]. In line with the former results, tibial fracture models treated with MPC showed the significant highest stiffness both with and without screw osteosynthesis. Those findings might be explained with the high intrinsic mechanical performance of the cement in the pure material tests and the seamless interdigitation to the spongiosa (Figure 10c and Figure 11a). Accordingly, it can be hypothesized that under increasing axial loading the specimens with MPC immediately resisted the loading forces, whereas in groups with a poorer interdigitation the bone substitute was shifted to a certain degree out of the drill channel, resulting in the lower stiffness of other groups. This theory is supported by a former study, in which the whole bone substitute Norian Drillable was pressed out of the drill channel under maximal loading [26]. Additionally, osteosynthesis significantly increased the stiffness of every group (Figure 9d); it can be concluded, as already done from the maximum load values, that a combination of bone substitute and osteosynthesis should always be performed.

Beyond the in vitro tests in this study, stress distribution of fractures and their stabilization methods can be analyzed by three-dimensional virtual models like the finite element analysis or Von Mises analysis [41,42,43]. Especially in dental surgery, computer-based simulation of prostheses and dental implants provide valuable information to guide the surgeon toward which implant position and size to choose [41,43]. In addition, those models can provide information about different mechanical properties of diverse materials of dental implants [41] and could be an interesting addition in a further study to the in vitro tests of the bone substitutes investigated in this study. 

Another crucial parameter in clinical scenarios like the treatment of tibial depression fractures is the viscosity of the applied cement system, as it might affect the interface between bone substitute and the adjacent spongiosa as well as the complete and precise filling of the defect. Thus, like the MPC demonstrated (Figure 7, Figure 8 and Figure 9), there might be a correlation between pure material testing and testing in the bone compound when the viscosity of the cement is appropriate. Concerning the limitations of this study, blocks of bone substitutes were only tested under uniaxial compression, despite in vivo bone substitutes being additionally exposed to more complex forces which consist of tensile, bending, and torsional components. Further, it is assumed that a fracture model with synthetic bones is limited in simulating physiological conditions, as no efforts were taken to consider the influence of soft-tissue and ingrowing bone into the implanted bone cements. Also, a storage with water-soaked gauzes around the Synbones^®^ may not correctly reflect in vivo humidity conditions. Bone cement in vivo is surrounded by a moist or aqueous environment immediately after injection, whereas in Synbones^®^ the inner surface of the drill channel is dry and contact with water may be limited to the cranial and caudal end of the cement body.

The biomechanical test set-up is similar to former studies in literature with regard to loading level, number of cycles, separating load-to-failure and cyclic testing, and concentrating on the main axial forces on the tibial plateau [28,44,45]. In contrast to separated load-to-failure and cyclic testing, McDonald et al. [46] designed a testing protocol with continuously increasing loading levels over a higher number of cycles overall. Compared with the aforementioned publication, the main interest in this study was more on the secondary displacement of the fracture fragment under loading approximated to the typical clinical postoperative loading conditions. By loading the stabilized fracture fragment directly with an indenter and by creating a resulting contact stress slightly above the values during gait, a more rigorous test set-up was performed with the focus on the lateral tibial plateau and the depressed articular fracture fragment. 

## 5. Conclusions

In conclusion, using the as-shown MPC within a tibial plateau depression fracture model is characterized by a high biomechanical stability in both pure static material tests and dynamic bone/cement interaction tests, though, at this point of time, one can only speculate about its assumingly high resorbability. The comparison between a commercial non-drillable (Graftys^®^ Quickset) and an in-house-prepared drillable CPC, both apatite-forming, when used with a screw osteosynthesis technique, demonstrated that a drillable bone substitute is favorable in terms of precise defect filling. This is not restricted to tibial head depression fractures, but conceivable for all complex fracture stabilization techniques where screws hinder a complete filling of the defect. In addition, this study implied the importance of an appropriate cement viscosity, as this enables the cement to keep its biomechanical stability shown in pure material tests when applied in the cement bone compound.

## Figures and Tables

**Figure 1 materials-12-01364-f001:**
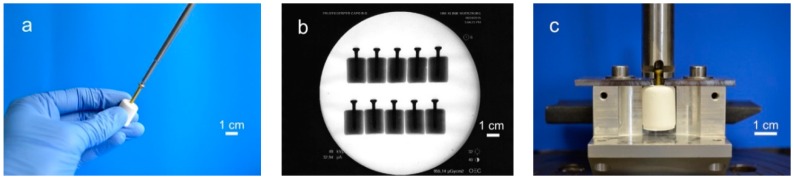
Screw pullout test setup. Workflow of specimen preparation for pullout tests: (**a**) In case of the drillable bone substitute, screws could be manually drilled into the cement. In all other cases screws were embedded in silicon rubber molds, in which cement was poured. (**b**) The cylindrical probes were then analyzed via X-rays to ensure equal screw placement, (**c**) prior to applying a tensile load with a custom-made test device for screw pullout.

**Figure 2 materials-12-01364-f002:**
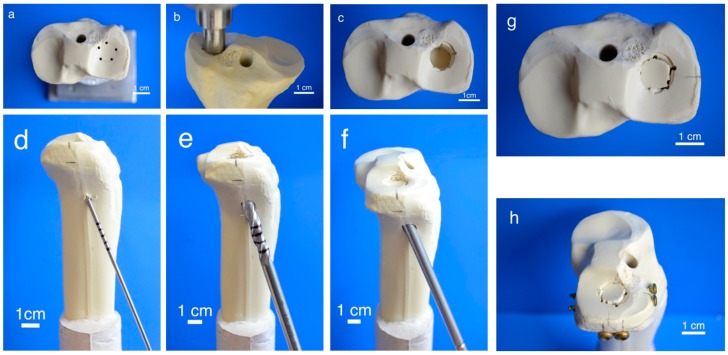
Fracture simulation, reduction, and additional screw osteosynthesis. Workflow of specimen preparation for the fracture model: Five determined breaking points (**a**) for the indenter (**b**) were set on the lateral tibial plateau to produce a pure impression fracture (**c**). After placing a K-wire under the depressed fragment (**d**), the lateral corticalis was opened with a drill (**e**), enabling placement of a cannulated ram (**f**) to compact subchondral spongiosa and restore a plane joint surface (**g**). In groups 5–7, fracture stabilization was complemented with a four-screw osteosynthesis using the jail technique (**h**).

**Figure 3 materials-12-01364-f003:**
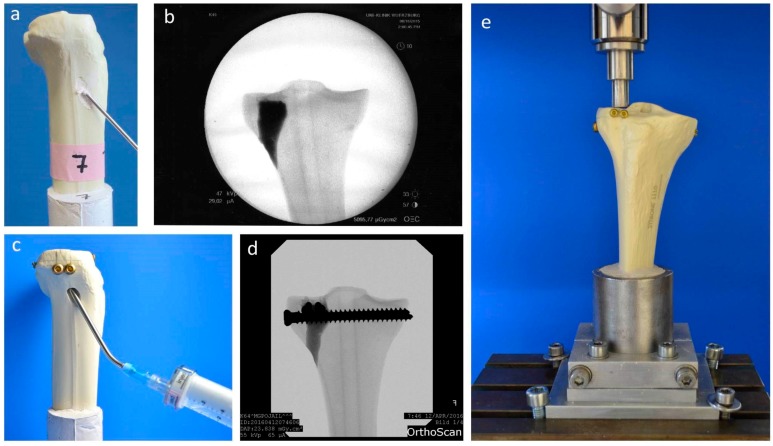
Application of bone cement and test set-up. Visualization of the different test groups: In groups 1–4, the remaining metaphyseal bone defect after fracture reduction was filled up with bone substitute only (**a**). In contrast, in groups 5–7, an additional screw osteosynthesis was set by the jail technique (**c**) and the filling of the defect was done before or after the osteosynthesis, depending on the drillability of the cement. To control the complete filling with cement and the position of the screws, specimens were examined by X-ray (**b**,**d**) before the biomechanical testing, where an axial load on the fixed fracture was performed (**e**).

**Figure 4 materials-12-01364-f004:**
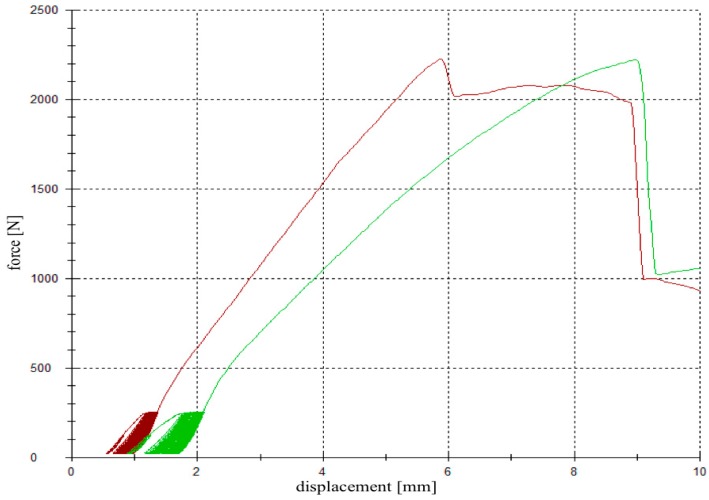
Load-displacement curve. A cyclic loading phase with 10 settling and 3000 measuring cycles was followed by a load-to-failure test. The red graph demonstrates the load-displacement curve of a specimen filled up with magnesium phosphate cement (MPC) (group 2), whereas the green graph corresponds to a specimen filled up with calcium phosphate cement (CPC) (group 3).

**Figure 5 materials-12-01364-f005:**
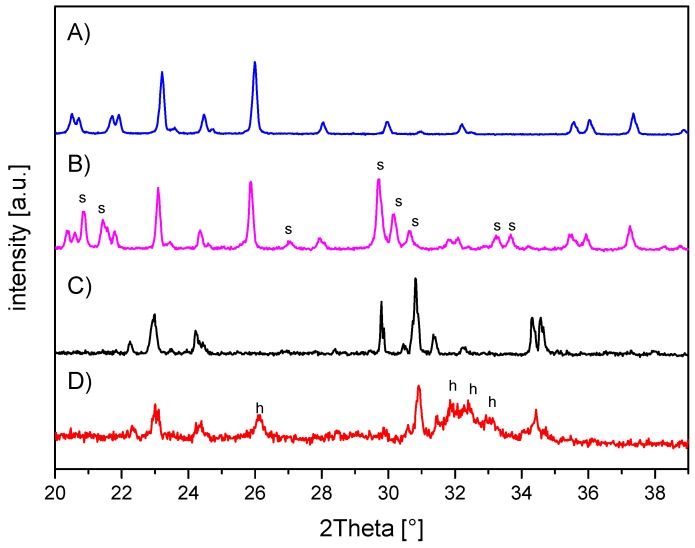
X-ray diffraction patterns of the in-house-prepared cements before and after setting at 37 °C for 24 h. (**A**) Magnesium phosphate cement raw powder; all peaks correspond to farringtonite Mg_3_(PO_4_)_2_ (PDF-No. 33-0876). (**B**) Magnesium phosphate cement after setting with additional struvite (s) phase (PDF-No. 03-0240). (**C**) α-tricalcium phosphate cement raw powder; all peaks correspond to α-TCP (PDF-No. 09-0348). (**D**) Dual setting α-TCP/HEMA cement after setting with additional hydroxyapatite (h) phase (PDF-No. 09-0432).

**Figure 6 materials-12-01364-f006:**
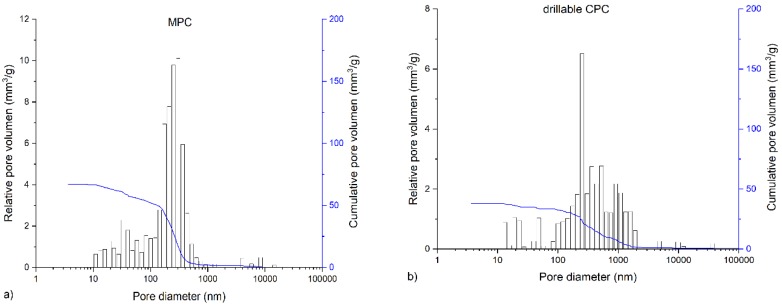
Graphs of mercury porosity analysis. Displayed are the graphs of the mercury porosity analysis for each bone substitute used in the study, (**a**) Magnesium phosphate cement; (**b**) Dual setting α-TCP/HEMA cement; (**c**) ChronOS^TM^ Inject; (**d**) Graftys^®^ Quickset. Relative pore volume (mm^3^/g) as well as cumulative pore volume (mm^3^/g) as a function of pore diameter are shown. Lowest cumulative porosity for in-house formulations and largest pores in ChronOS^TM^ were determined: An equivalent low cumulative pore volume was found for the drillable CPC and MPC, whereas the commercial formulations showed distinctly higher cumulative pore volumes. The highest average pore volume and median pore volume were determined for ChronOS^TM^ Inject.

**Figure 7 materials-12-01364-f007:**
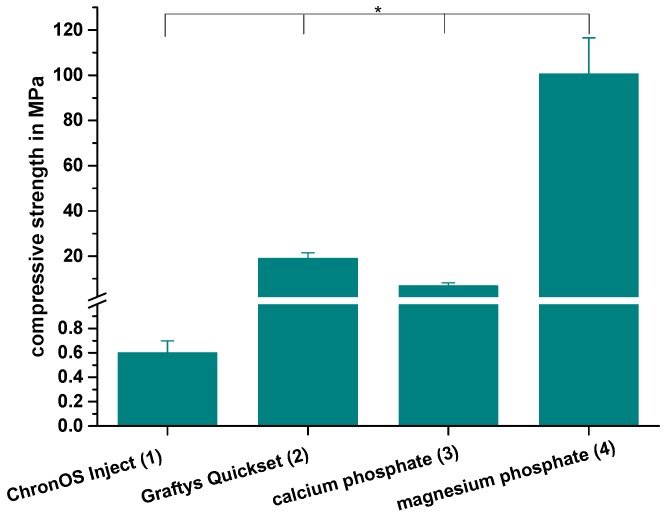
Compressive strength. Significant highest compressive strength for the MPC: Compressive strength of cuboidal samples from commercial ChronOS^TM^ Inject (group 1), Graftys^®^ Quickset (group 2) and in-house-prepared dual setting calcium phosphate (group 3) and magnesium phosphate (group 4) after 24 h of setting in water at 37 °C. All differences between the single groups were significant (groups 1,2; groups 1,3; groups 1,4; groups 2,3; groups 2,4; groups 3,4). Significant differences (*) were set at *p* < 0.01 each.

**Figure 8 materials-12-01364-f008:**
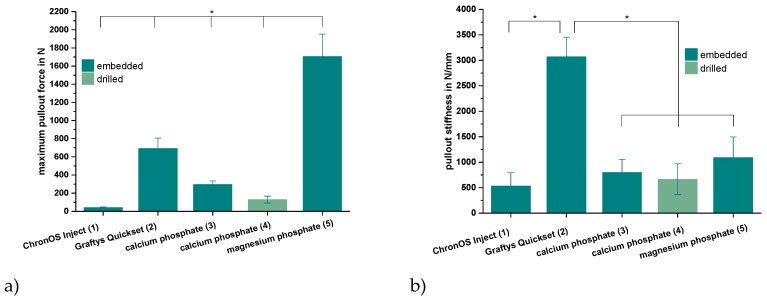
Maximum pullout force and pullout stiffness. Highest pullout force for the MPC and highest pullout stiffness for Graftys^®^: Maximum force (**a**) and stiffness (**b**) while pulling out cortical screws from a cylindrical matrix based on commercial ChronOS^TM^ Inject (group 1), Graftys^®^ Quickset (group 2) and in-house-prepared dual setting calcium phosphate (groups 3,4) and magnesium phosphate (group 5) after 24 h of setting in water at 37 °C. Only in the case of the dual setting calcium phosphate, the cement matrix sustained tapping and drilling (group 4), whilst in all other cases, the screws had to be embedded in the unhardened matrix. All differences between the single groups were significant for the maximum pullout force (**a**). Graftys^®^ Quickset showed the significantly highest stiffness values of all experimental groups. Significant differences (*) were set at *p* < 0.01 each.

**Figure 9 materials-12-01364-f009:**
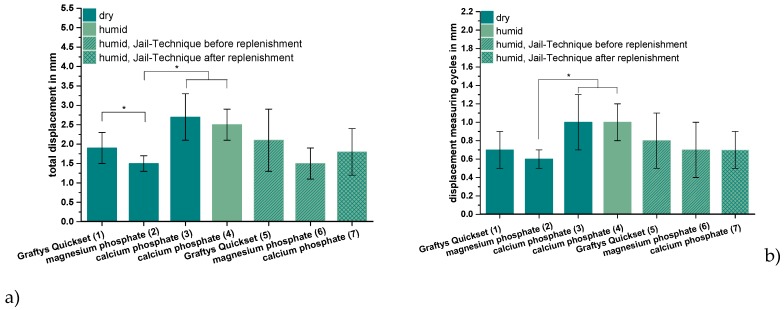
Displacement, maximum load, and stiffness of the dynamic tests. Displacement of the augmented tibial fractures during whole test protocol, including (**a**) settling cycles and (**b**) during the measuring cycles only, (**c**) maximum load, and (**d**) stiffness of the constructs after 24 h of setting in dry (groups 1–3) and humid conditions (groups 4–7) at 37 °C, are shown. Augmentation was based on commercial Graftys^®^ Quickset (groups 1,5) and in-house-prepared dual setting calcium phosphate (groups 3,4,7) and magnesium phosphate (groups 2,6). Pure augmentation was done in groups 1–4, whereas, in groups 5 and 6, a jail technique was applied additionally before and, in group 7, after replenishment, depending on the drillability of the cement. Pure augmentation with magnesium phosphate showed significant lower values compared to other groups in terms of both total displacement and displacement in measuring cycles only (groups 2,3; groups 2,4; significant difference between group 1,2 only for total displacement) (**a**,**b**). Regarding the maximum load, the humid CPC revealed a significant higher maximum load than the CPC stored under dry conditions (groups 3,4). All differences between the groups with bone substitute only (groups 1–4) and the groups with the screws in the jail technique (groups 5–7) were significant (**c**). Concerning the stiffness, the MPC exhibited the significant highest values in the group with bone substitute only (groups 1–4) and in the group with osteosynthesis (groups 5–7). The stiffness was calculated as a measure for the ascending slope during the elastic deformation (**d**). Significant differences (*) were set at *p* < 0.01 each.

**Figure 10 materials-12-01364-f010:**
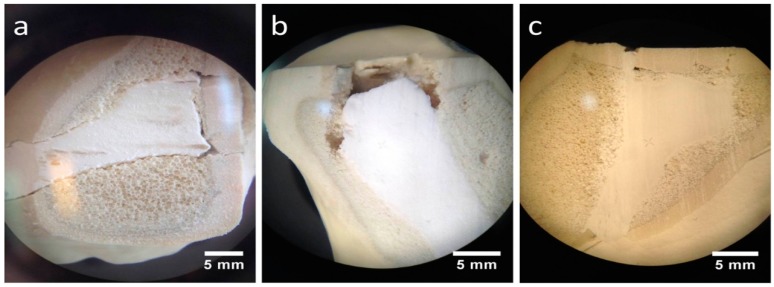
Stereomicroscopic images. Better interdigitation of the humid CPC and seamless interdigitation of the MPC: Regarding augmented tibial head depression fractures under a stereomicroscope, a low interdigitation with a visible gap to the spongiosa could be seen for the CPC when stored under dry conditions (**a**). In contrast, under humid storage conditions, the interdigitation for the same cement seemed to improve remarkably, showing no gap (**b**). A full interdigitation to the nearby spongiosa up to the reduced fracture fragment was detected for the MPC (**c**).

**Figure 11 materials-12-01364-f011:**
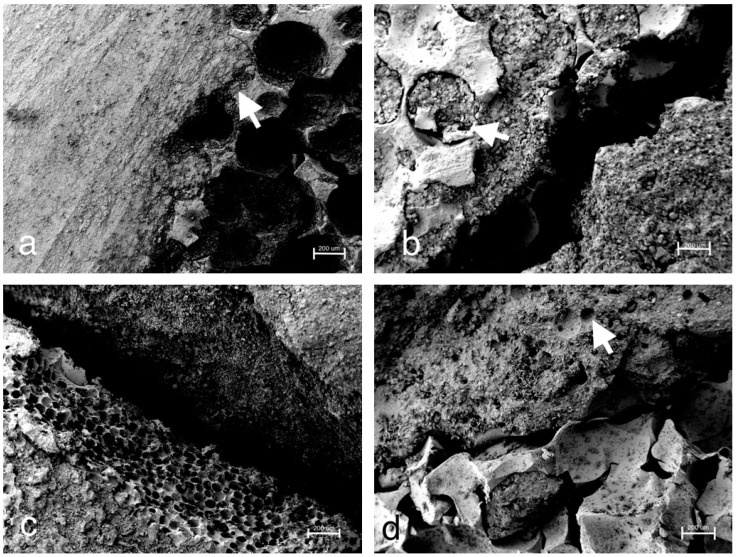
SEM images of the cement–bone interface. Smooth interdigitation of the MPC and confirmation of stereomicroscopic findings: As already revealed on the stereomicroscopic images, the MPC demonstrated a seamless filling of the adjacent spongiosa cavities at the cement–bone junction (arrowhead in (**a**)). For the drillable CPC stored under humid conditions, in contrast to the stereomicroscopic images, a gap to the adjacent spongiosa became visible (**b**). However, a filling of nearby spongiosa cavities could be detected as well (arrowhead in (**b**)). During the preparation process for SEM, the humid cement was dried necessarily in a desiccator. Congruently, the SEM images for the drillable CPC stored under dry conditions revealed a gap (**c**). Graftys^®^ Quickset demonstrated a close connection to the adjacent spongiosa (**d**). Moreover, the macroporous and mesoporous structures of the bone substitute can be visualized (arrowhead in (**d**)).

**Table 1 materials-12-01364-t001:** Bone cements; overview of the commercial and in-house-prepared cement systems used for biomechanical analysis in this study.

	Commercial		In-House	
Label/trade name	ChronOS^TM^ Inject	Graftys^®^ Quickset	calcium phosphate	magnesium phosphate
Product	brushite	calcium-deficient hydroxyapatite	calcium-deficient hydroxyapatite	struvite
Biodegradability	yes	for years	no	yes
Drillability	no	no	yes	no

**Table 2 materials-12-01364-t002:** Experimental groups; display of the seven groups for biomechanical analysis of tibial head fractures using osteosynthesis techniques and storage conditions.

Group	Bone Substitue	Osteosynthesis	Storage	n
1	Graftys^®^ Quickset		dry	9
2	magnesium phosphate		dry	9
3	calcium phosphate		dry	9
4	calcium phosphate		humid	9
5	Graftys^®^ Quickset	4 screws jail before replenishment	humid	9
6	magnesium phosphate	4 screws jail before replenishment	humid	9
7	calcium phosphate	4 screws jail after replenishment	humid	9

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
