# Peer review of "Biomechanical Evaluation of Promising Different Bone Substitutes in a Clinically Relevant Test Set-Up"

_materials, 2019, doi:10.3390/ma12091364_

Round 1
Reviewer 1 Report
This paper presents results about biomechanical tests of 4 types of bone grafts substitutes.
This well-written paper will benefit from some modifications/clarification such as:
Abstract:
Page 1 line 16: remove “therefore.”
Page 1 line 17: remove “from the first time in literature.”
Page 1 line 20: replace “analyzed” with “investigated.”
Page 1 line 24: remove “it seems.”
Page 1 line 28: remove “Probably.”
Introduction: Too long and should focus more on the characteristics of each material tested. The rationale of the hypothesis is not clear.
Line 41 sentence does not make sense in the context; please rephrase
Line 53 the degradability is often related to the crystallinity of HA, please elaborate on this parameter.
Line 62: please use trabecular instead of spongiosa
Line 63: please start a new paragraph at To improve …
Line 72 and 73: please elaborate on the mechanical properties and the solubility of the MPC, and what are the parameters that provide these values.
Line 74: please elaborate on the differences between the 4 and how do you expect that each will degrade.
Material and methods: please use subsection 2.1 and 2.2
Line 121: please provide more information about the mercury porosity analysis. More specifically, what was the imaging software used?
Figures: please elaborate each caption and provide the key message in them.
Figure 1: scale is missing
Figure 2: scale is missing
Mechanical test, please provide more details on the data analysis of the mechanical tests. For instance, you provide data on stiffness and show the force-displacement curve.
Could you please explain how you obtain the data.
Furthermore, did you calculate the energies dissipated as well, please provide this information.
Statistics :
Line 221 Please provide the data of the power analysis.
SEM images analysis appear only in the results section; please integrate this in the MM part. ( same for the stereomicroscopic image)
Figure 10: a, b, c, and d are missing in the figure itself.
Discussion:
Line 456-458: please elaborate on the crystal properties you are referring to.
Line 479-481: please clarify this in the results section.
Author Response
We thank the reviewer for his valuable commentaries.
Abstract:
Page 1 line 16: remove “therefore.”
Page 1 line 17: remove “from the first time in literature.”
Page 1 line 20: replace “analyzed” with “investigated.”
Page 1 line 24: remove “it seems.”
Page 1 line 28: remove “Probably.”
Commentary: We corrected the points in the text.
Introduction: Too long and should focus more on the characteristics of each material tested. The rationale of the hypothesis is not clear.
Commentary: We shortened the introduction and rephrased the hypothesis.
Line 41 sentence does not make sense in the context; please rephrase
Commentary: We rephrased the sentence.
Line 53 the degradability is often related to the crystallinity of HA, please elaborate on this parameter.
Commentary: We agree that this has an influence on degradability, however we feel that this point is very special and out of focus of the study. Instead, we have generally explained the degradation behaviour of the different cements in more detail.
Line 62: please use trabecular instead of spongiosa
Line 63: please start a new paragraph at To improve …
Commentary: We corrected both points in the text.
Line 72 and 73: please elaborate on the mechanical properties and the solubility of the MPC, and what are the parameters that provide these values.
Line 74: please elaborate on the differences between the 4 and how do you expect that each will degrade.
Commentary: We have elaborated this part with a focus on degradation and mechanical performance.
Material and methods: please use subsection 2.1 and 2.2
Commentary: We divided the M&M section in subsections and added the used imaging software used.
Line 121: please provide more information about the mercury porosity analysis. More specifically, what was the imaging software used?
Commentary: We have added the relevant information. Please note that there is no imaging software necessary for this method. Pore sizes and their distribution are calculated on base of the pressure which is necessary to press mercury into the samples.
Figures: please elaborate each caption and provide the key message in them.
Figure 1: scale is missing
Figure 2: scale is missing
Commentary: We have redone the captions and added the scales.
Mechanical test, please provide more details on the data analysis of the mechanical tests. For instance, you provide data on stiffness and show the force-displacement curve.
Could you please explain how you obtain the data. Furthermore, did you calculate the energies dissipated as well, please provide this information.
Commentary: We added a section about the data collection in detail in the text.
Statistics :
Line 221 Please provide the data of the power analysis.
Commentary: We added the data of the power analysis done by the mathematical institute of our university in detail.
SEM images analysis appear only in the results section; please integrate this in the MM part. (same for the stereomicroscopic image)
Figure 10: a, b, c, and d are missing in the figure itself.
Commentary: We integrated the images in the M&M part and corrected Figure 10.
Discussion:
Line 456-458: please elaborate on the crystal properties you are referring to.
Line 479-481: please clarify this in the results section.
Commentary: As mentioned above, we did not add more details about the crystal properties of the HA cements and have therefore rewritten this part. We believe that the increasing strength is a result of a proceeding setting reaction, which is not yet finished after 24 h for the in house prepared apatite cement.
Reviewer 2 Report
The manuscript is very interesting to scientists working on clinical practice or on fundamental properties of bone filling materials. From the biomechanical aspect this is a comprehensive study. However since the host journal is “Materials” it should include also the structural and physicochemical properties of the used materials. For example XRD graphs of raw materials and hardened cements, SEM images or other spectroscopic characterization. My suggestion is acceptance after major revision.
Author Response
The manuscript is very interesting to scientists working on clinical practice or on fundamental properties of bone filling materials. From the biomechanical aspect this is a comprehensive study. However since the host journal is “Materials” it should include also the structural and physicochemical properties of the used materials. For example XRD graphs of raw materials and hardened cements, SEM images or other spectroscopic characterization. My suggestion is acceptance after major revision.
Commentary: We thank the reviewer for this valuable commentary to the reasonable addition to the basic material section. We performed the additional tests and added the XRD graphs for the in house prepared cements to the manuscript.
Reviewer 3 Report
The paper is well done and structured.
The topic fits within the MATERIALS aim and scope and it could be attractive for the Journal Readers. The figures are impressive.
Just minor concerns about the discussion. Some other method for testing the stress over bone should be added and compared like virtual model as FEM and Von Mises analysis.
Some suggestion or paper that could be added in the discussion section
Shultz, T.R.; Blaha, J.D.; Gruen, T.A.; Norman, T.L. Cortical bone viscoelasticity and fixation strength of press-fit femoral stems: Finite element model. J. Biomech. Eng. 2006, 128, 7–12.
Cicciù, M., Cervino, G., Milone, D., & Risitano, G. (2018). FEM Investigation of the Stress Distribution over Mandibular Bone Due to Screwed Overdenture Positioned on Dental Implants. Materials (Basel, Switzerland), 11(9), 1512. doi:10.3390/ma11091512
Bramanti, E., Cervino, G., Lauritano, F., et al.. (2017). FEM and Von Mises Analysis on Prosthetic Crowns Structural Elements: Evaluation of Different Applied Materials. The Scientific World Journal, 2017, 1029574. doi:10.1155/2017/1029574
Author Response
The paper is well done and structured.
The topic fits within the MATERIALS aim and scope and it could be attractive for the Journal Readers. The figures are impressive.
Just minor concerns about the discussion. Some other method for testing the stress over bone should be added and compared like virtual model as FEM and Von Mises analysis.
Some suggestion or paper that could be added in the discussion section
Shultz, T.R.; Blaha, J.D.; Gruen, T.A.; Norman, T.L. Cortical bone viscoelasticity and fixation strength of press-fit femoral stems: Finite element model. J. Biomech. Eng. 2006, 128, 7–12.
Cicciù, M., Cervino, G., Milone, D., & Risitano, G. (2018). FEM Investigation of the Stress Distribution over Mandibular Bone Due to Screwed Overdenture Positioned on Dental Implants. Materials (Basel, Switzerland), 11(9), 1512. doi:10.3390/ma11091512
Bramanti, E., Cervino, G., Lauritano, F., et al.. (2017). FEM and Von Mises Analysis on Prosthetic Crowns Structural Elements: Evaluation of Different Applied Materials. The Scientific World Journal, 2017, 1029574. doi:10.1155/2017/1029574
Commentary: Thank you for the positive review and the supplementary notes to the discussion. We added a section to the computer-based models and the reasonable addition to the tests in this manuscript.
Round 2
Reviewer 2 Report
The manuscript has been improved and my suggestion is acceptance for publication